# Transcriptomic Profiling of Meat Quality Traits of Skeletal Muscles of the Chinese Indigenous Huai Pig and Duroc Pig

**DOI:** 10.3390/genes14081548

**Published:** 2023-07-28

**Authors:** Xiaojin Li, Liangyue Lu, Xinwei Tong, Ruidong Li, Erhui Jin, Man Ren, Yafei Gao, Youfang Gu, Shenghe Li

**Affiliations:** 1College of Animal Science, Anhui Science and Technology University, Chuzhou 233100, China; lixj@ahstu.edu.cn (X.L.); t1125539377@163.com (L.L.); t1125539377@126.com (X.T.); m17563400413@163.com (R.L.); jineh@ahstu.edu.cn (E.J.); renm@ahstu.edu.cn (M.R.); 2Anhui Province Key Laboratory of Animal Nutritional Regulation and Health, Chuzhou 233100, China; 3Key Laboratory of Quality and Safety Control for Pork, Ministry of Agriculture and Rural, No. 9, Chuzhou 233100, China; gaoyafling@163.com

**Keywords:** Huai pig, IMF, transcriptome analysis, longissimus dorsi muscle

## Abstract

The Huai pig is a well-known indigenous pig breed in China. The main advantages of Huai pigs over Western commercial pig breeds include a high intramuscular fat (IMF) content and good meat quality. There are significant differences in the meat quality traits of the same muscle part or different muscle parts of the same variety. To investigate the potential genetic mechanism underlying the meat quality differences in different pig breeds or muscle groups, longissimus dorsi (LD), psoas major (PM), and biceps femoris (BF) muscle tissues were collected from two pig breeds (Huai and Duroc). There were significant differences in meat quality traits and amino acid content. We assessed the muscle transcriptomic profiles using high-throughput RNA sequencing. The IMF content in the LD, PM, and BF muscles of Huai pigs was significantly higher than that in Duroc pigs (*p* < 0.05). Similarly, the content of flavor amino acids in the three muscle groups was significantly higher in Huai pigs than that in Duroc pigs (*p* < 0.05). We identified 175, 110, and 86 differentially expressed genes (DEGs) between the LD, PM, and BF muscles of the Huai and Duroc pigs, respectively. The DEGs of the different pig breeds and muscle regions were significantly enriched in the biological processes and signaling pathways related to muscle fiber type, IMF deposition, lipid metabolism, PPAR signaling, cAMP signaling, amino acid metabolism, and ECM–receptor interaction. Our findings might help improve pork yield by using the obtained DEGs for marker-assisted selection and providing a theoretical reference for evaluating and improving pork quality.

## 1. Introduction

Skeletal muscles are the main meat-producing tissue of pigs and have a unique cellular structure. Skeletal muscle fibers are the largest cells in mammals, accounting for 75–90% of the skeletal muscle tissue [1]. The number, size, and type of muscle fibers determine the basic characteristics of a muscle and its yield. Muscle fiber characteristics are closely related to pork meat color, pH, muscle tenderness, and intramuscular fat (IMF) content [2,3]. The IMF content affects muscle tenderness and flavor, with higher IMF content being associated with higher pork quality [4]. The type of muscle fiber is a key factor that affects pork quality. Muscle growth after birth mainly depends on the thickening and transformation of muscle fibers [5,6].

Pork quality is a quantitative trait controlled by micro-efficacious polygenes and belongs to the medium and high heritability traits that play a decisive role in varieties [7]. Meat quality traits differ significantly between breeds, especially between domestic and foreign breeds. Imported pig breeds, such as the Duroc, have a high growth rate, high glycolytic fiber content, and low IMF content [8]. Chinese indigenous pig breeds, such as the Huai, are characterized by high IMF content; high diameter of I, IIa, and IIb muscle fibers; and a high content of oxidized fibers [9,10]. Huang et al. [11] reported that Bama Xiang pigs have a high longissimus dorsi (LD) type I muscle fiber content, whereas Landrace pigs have a high glycolytic IIb muscle fiber content. Muscle fiber composition is a key factor causing differences in meat quality between Bama Xiang and Landrace pigs. The meat quality traits of different muscle groups within the same variety of pig are also significantly different. Moreover, the muscle fiber diameter and area of the LD muscle are significantly higher than those of the psoas major (PM) muscle [12,13]. Furthermore, the muscle fiber composition ratio of the same muscle group of the same variety or different muscle groups of the same variety affects pork meat quality. A better understanding of the genetic mechanisms underlying muscle fiber types is important for improving livestock and poultry meat quality.

Muscle growth is a complex physiological process, and a series of genes and other factors are important for accurately regulating it [14]. Transcriptomic studies using high-throughput sequencing technology have identified several transcription factors affecting pig muscle growth and gene regulatory networks regulating muscle production by comparing Chinese and foreign pig breeds. Xu et al. [15] compared the transcriptome data of the LD muscle of Yorkshire and Wei pigs, identified 717 differentially expressed genes (DEGs), and obtained four candidate genes related to fat metabolism (*FABP3*, *PDK4*, *ACSL1*, and *UCP3*). Shang et al. [16] conducted transcriptomic and proteomic sequencing on Tibetan, Wujin, and White pigs. They discovered that 20 genes related to muscle fiber formation, including *CRYAB*, *FSCN1*, and *MAPK12*, may play an important role in determining the postnatal growth rate of pigs. Zhao et al. [17] compared the LD muscles of Tongcheng and Yorkshire pig embryos from 30 days to 5 weeks after birth and discovered that DEGs regulating muscle development showed variety-specific differential expression patterns.

In this study, transcriptomic sequencing technology was used to analyze the meat quality traits of different muscle groups from Huai and Duroc pigs to further elucidate the differences in molecular regulatory mechanisms between breeds and muscle groups. Our data could provide an important reference for subsequent improvements in livestock and poultry meat quality as well as muscle production.

## 2. Materials and Methods

All experimental pigs were handled in strict accordance with the good animal practices of the People’s Republic of China Ethical Procedures and Guidelines for Animals and were approved by the Animal Management and Ethics Committee of Anhui University of Science and Technology (license number 2019-002).

### 2.1. Animal Sampling and Meat Composition Measurements

Huai and Duroc pigs were provided by the Anhui Haoxiang Agriculture and Animal Husbandry Co., Ltd., Bozhou, China. Notably, three healthy Huai and Duroc pigs were fed under the same conditions and slaughtered at 180 days of age. LD, PM, and biceps femoris (BF) muscle tissues between the left penultimate third and fourth ribs were collected. Some samples were used for determining meat quality traits. The remaining samples were frozen in liquid nitrogen immediately after being packaged in a freezing tube and then transferred to the laboratory for storage at −80 °C until used. The IMF content was measured using Soxhlet petroleum ether extraction [18].

The types and contents of amino acids in the LD, PM, and BF muscles of the Huai and Duroc pigs were determined according to a previously described method in *Determination of Amino Acids in National Standard Food Safety* (GB 5009.124-2016) [19].

### 2.2. Total RNA Extraction and Illumina Sequencing

Total RNA was extracted from muscle tissues using the TRIzol reagent (Thermo Fisher Scientific, Carlsbad, CA, USA). The concentration, purity, and integrity of total RNA were determined using a NanoDrop ND-2000 spectrophotometer (Thermo Fisher Scientific, Wilmington, DE, USA) and an Agilent Bioanalyzer 2100 (Agilent Technologies, Santa Clara, CA, USA), using the RNAClean XP Kit (Beckman Coulter, Inc., Kraemer Boulevard Brea, CA, USA) and the RNase-Free DNase Set (QIAGEN, GmBH, Frankfurt, Germany) for purification.

A transcriptome library was constructed by Shanghai Biotechnology Corporation (Shanghai, China). Referring to the NEBNext Ultra RNA Library Prep Kit for Illumina (New England Biolabs, Ipswich, MA, USA), the mRNA was purified by adsorption with Oligo (dT) magnetic beads, and fragmentation buffer was added to break the mRNA into short fragments. Subsequently, a cDNA strand was synthesized through reverse transcription using six-base random hexamers as a template, and a two-strand cDNA was synthesized by adding buffer, dNTPs, and DNA polymerase I. Thereafter, the double-stranded cDNA was purified using AMPure XP beads. The purified double-stranded cDNA was repaired at the end, and A as well as splice were added. AMPure XP beads were used for fragment size selection of double-stranded cDNA and PCR amplification was performed to construct cDNA library. Finally, all libraries were sequenced.

The raw data obtained were pretreated using Fastx, and low-quality reads, such as those with only 5′ and 3′ end joints, poly-N, or low overall quality, were removed to obtain clean reads. Using the Hisat2 (version 2.0.4) software [20], clean sequences were compared to the Sus scrofa reference genome (Sscrofa11.1) sequences to obtain the location information on the reference genome or gene to obtain mapped reads. Gene quantification of mapping results was performed using the StringTie (version:1.3.0) software [21]. The gene expression level is positively correlated with the abundance of transcripts, and the FPKM value is used in transcriptome sequencing to measure the expression level of each gene in the sample [18].

The DESeq software package in R was used for differential gene expression analyses. Significant DEGs are identified between any two groups based on the following thresholds: log2lfold-changel ≥ 1 and padj-value ≤ 0.05 (Benjamini and Hochberg methods) [22].

### 2.3. Functional Annotation

Gene Ontology (GO) and Kyoto Encyclopedia of Genes and Genomes (KEGG) analyses of DEGs were performed using the Bioconductor package clusterProfiler [23]. The obtained *p* values were adjusted according to Benjamini and Hochberg methods, and padj values ≤ 0.05 were considered significant.

### 2.4. Real-Time Quantitative PCR

The differential expression patterns of genes detected in the transcriptome data were verified using RT-qPCR analyses. Notably, 21 DEGs were associated with muscle development.

Total RNA (50 µg) was extracted from LD, leg, and waist muscle tissue samples from different pig breeds using a TRIzol kit (Thermo Fisher, Shanghai, China). qPCR was performed on SYBR Green I-treated samples using Roche 96. Each qPCR assay was performed in a 20 µL volume consisting of 2 µL DNA template, 0.4 µL up- and downstream primers, 10 µL SYBR Premix Ex Taq II reverse transcriptase (Thermo Fisher, Shanghai, China), and 7.2 µL RNase-free ddH_2_O. The fluorescence quantitative results were calculated using the 2^−ΔΔCt^ method, where ΔCt = Ct (target gene) − Ct (internal reference gene). Statistical analyses were performed using IBM SPASS 20.0 software, and the results are expressed as mean ± SD. The primer sequences are listed in Appendix A. All samples were tested three times, and negative controls were set up for each test.

## 3. Results

### 3.1. Muscle Quality Analysis

The meat quality traits of the LD, PM, and BF muscles in the Huai and Duroc pigs were assessed. The a-value, hydraulic power, and elasticity of the LD, PM, and BF muscles of the Huai pigs were significantly higher than those of the Duroc pigs (*p* < 0.01). In contrast, the L-value, water loss, and cooking loss of the Huai pigs were significantly lower than those of the Duroc pigs (*p* < 0.01). The water content was significantly higher in Huai pigs than in Duroc pigs (*p* < 0.05). Furthermore, the IMF content in the LD and PM of Huai pigs was significantly higher than that in Duroc pigs (*p* < 0.01). The IMF of the BF in Huai pigs was significantly higher than that in Duroc pigs (*p* < 0.05) (Table 1, Table 2 and Table 3).

### 3.2. Amino Acid Composition and Content

The amino acid contents in the LD, PM, and BF muscles of Huai and Duroc pigs were measured. The levels of flavor amino acids (glutamic acid, aspartic acid, phenylalanine, alanine, glycine, and tyrosine) in the LD, PM, and BF muscles of Huai pigs were significantly higher than those in Duroc pigs (*p* < 0.05) (Table 4, Table 5 and Table 6).

### 3.3. Summary of RNA-Seq Results

Table 7 summarizes the RNA-seq results. The unique ratio following mapping to the reference genome (*Sus scrofa*, version 11.1) ranged between 84.19% and 87.77% with good alignment. Pearson’s correlation coefficient (R^2^) was used as an evaluation index for biological replicate correlation. The R^2^ for sample expression was 96.0% for HB1, HB2, and HB3; 93.0–96.0% for DLB1, DLB2, and DLB3; 95.0–96.0% for HY1, HY2, and HY3; 93.0–96.0% for DLY1, DLY2, and DLY3; 92.0–95.0% for HT1, HT2, and HT3; and 93.0–94.0% for DLT1, DLT2, and DLT3; thus confirming the robustness of the biological replicates and the reliability of the RNA-seq results (Figure 1).

### 3.4. Differential Expression Analysis

The edgeR [24] was used to identify DEGs. *p* < 0.05 and |log2FC| ≥ 1 were used as criteria to screen the DEGs between HB vs. DLB, HY vs. DLY, and HT vs. DLT. A total of 175, 110, and 86 DEGs were identified for the comparison between HB and DLB, HY and DLY, and HT and DLT, respectively. Of these DEGs, 76, 46, and 45 were upregulated and 99, 64, and 41 were downregulated, respectively (Figure 2a–d).

Venn diagram analysis of DEGs with an FPKM >1 was performed, and 9440 differentially co-expressed genes were screened from three different muscle tissues (HB vs. DLB, HY vs. DLY, and HT vs. DLT) of Huai and Duroc pigs. The resulting heatmap indicates that the three samples from different muscle tissues of Huai and Duroc pigs were in the same cluster. At least twice as many DEGs were observed in the different muscle tissues of Huai and Duroc pigs, which further confirmed the sampling accuracy and reliability in this study (Figure 3a–d).

### 3.5. Functional Enrichment Analysis

Functional enrichment analysis was performed for the three groups of DEGs screened using the GO database. The GO enrichment analysis was divided into three categories: biological processes, cellular components, and molecular functions. Figure 4 shows the 30 most significantly enriched GO terms for each group. For HB vs. DLB, 164 GO items were significantly enriched (*p* < 0.05), including “actomyosin”, “heat shock protein binding”, “steroid hormone-mediated signaling pathway”, and “regulation of lipid metabolic process” (Figure 4a). For HY vs. DLY, 107 GO items were significantly enriched (*p* < 0.05), including “regulation of calcium ion transport”, “steroid hormone-mediated signaling pathway”, “fat cell differentiation”, “G-protein coupled receptor binding”, and “muscle tissue development” (Figure 4b). For HT vs. DLT, 48 GO items were significantly enriched (*p* < 0.05), including “myofibril assembly”, “neuromuscular process”, “striated muscle cell development”, and “muscle cell development” (Figure 4c).

The KEGG analysis of the DEGs identified significantly enriched pathways at different developmental stages (Figure 5). For HB vs. DLB, 18 pathways were significantly enriched (*p* < 0.05), including the “glycine, serine, and threonine metabolism”, “arginine and proline metabolism”, and “ECM–receptor interaction” signaling pathways (Figure 5a). For HY vs. DLY, 23 pathways were significantly enriched (*p* < 0.05). These included the “alanine, aspartate, and glutamate metabolism”, “PPAR signaling pathway”, and “RIG-I-like receptor signaling pathway” signaling pathways (Figure 5b). For HT vs. DLT, 15 pathways were significantly enriched (*p* < 0.05), including the “hypertrophic cardiomyopathy (HCM)”, “cardiac muscle contraction”, and “TNF signaling” pathways (Figure 5c).

### 3.6. Validation of RNA-Seq Data Using qRT-PCR

DEGs were randomly selected, and qRT-PCR was used to verify the DEGs between the Huai and Duroc pigs. The qRT-PCR results were consistent with our RNA-seq data (Figure 6). The linear regression between DEGs data obtained from the qRT-PCR and RNA-seq results shows a high correlation (R^2^ = 0.8963), indicating the reliability of our RNA-seq results.

## 4. Discussion

Water and IMF, the most common traits of pork, contribute to pork juiciness, and the IMF content is highly correlated with flavor. The oxidized muscle fibers (MyHC I and MyHC Пa) had lower glycogen content, higher myoglobin and phospholipid content, and higher aerobic metabolic capacity than those of MyHC IIb. The metabolic activity and contractile characteristics of MyHC IIx were between those of the oxidized and glycolytic muscle fibers. In addition, the expression levels of MyHC I, MyHC IIa, and MyHC IIx were positively correlated with pH, flesh color, marblework, and IMF content, whereas MyHC IIb expression was inversely correlated with these parameters [25,26]. When the proportion of oxidized muscle fibers is high, the muscle fiber diameter is smaller, the flesh color and marbling scores are higher, the drip loss is reduced, and the IMF content as well as water retention are higher [27,28]. In this study, the L-value, drip loss, cooking loss, and hardness of the LD, PM, and BF muscles of the Huai pigs were shown to be significantly lower than those of the Duroc pigs (*p* < 0.05). In contrast, the hydraulic, elasticity, and IMF contents of the Huai pigs were significantly higher than those of the Duroc pigs (*p* < 0.05), which is consistent with results from previous studies [29,30]. Therefore, the meat quality of Huai pigs is better than that of Duroc pigs, and the different types of muscle fibers directly affect pork quality.

For same breed animals, the type of muscle fibers in different muscle groups also varies greatly. Most muscles are composed of both oxidized and glycolytic fibers. Deep muscles have a higher proportion of oxidized fibers than shallow muscles. The BF muscle is the largest muscle tissue, with a higher oxidation capacity and lower drip loss than the LD muscle. In addition, there are important differences in IMF content between the two muscles [31]. In pigs, the LD and BF are mainly composed of fast glycolytic fibers. The PM is mainly composed of rapidly oxidizing and intermediate fibers; therefore, the proportion of oxidized muscle fibers in the PM is significantly higher than that in the LD and BF muscles [32,33]. Furthermore, the IMF content in the PM is significantly higher than that in the LD and BF muscles. We also discovered that the IMF content of the PM muscle of Huai and Duroc pigs was significantly higher than that of the LD and BF muscles, respectively.

Proteins are synthesized from amino acids, and the category as well as content of amino acids directly affect the nutritional value and flavor of pig meat. Therefore, small amino acids are strongly related to the quality and flavor of pig meat. Flavor is a major determinant of meat quality and consumer purchasing decision. Amino acids that can present a special flavor are called flavor amino acids and include glutamic acid, aspartic acid, phenylalanine, alanine, glycine, and tyrosine. The content of these flavor amino acids has been suggested to be particularly important in the quality and taste of pork [34,35]. Glutamic acid forms umami, buffering acid, and base taste flavors and plays an important role in flavor amino acids [36]. Fatty acids, phospholipids, and flavor amino acids affect muscle flavor traits [37]. The content of flavor substances in oxidized muscle tissue is higher than that in glycolytic muscle tissue, which further confirms that muscles with a higher proportion of oxidized muscle fibers have a preferable flavor [38,39]. The content of flavor amino acids in the LD and BF muscles of the Huai pigs was significantly higher than that in the Duroc pigs (*p* < 0.05), which is consistent with previous results [19,40]. Therefore, the amino acid content might represent the main factor that causes differences in the flavor, odor, and meat quality between the Huai and Duroc pigs.

Animal muscle development is a complex process involving multiple coregulating genes and signaling pathways. GO enrichment analysis of the DEGs from different muscle groups of Huai and Duroc pigs revealed the biological processes closely related to muscle development, IMF, and lipid metabolism, including muscle tissue development, striated muscle cell differentiation, lipid metabolism regulation, and myofibril formation. Notably, seven DEGs (*CSRP3*, *LMOD2*, *NR3C2*, *CXCL10*, *ACTC1*, *MYH10*, and *SPTLC3*) related to meat quality, muscle development, and lipid metabolism were identified during these significantly enriched biological processes. Compared to the levels in Duroc pigs, *CSRP3*, *LMOD2*, and *NR3C2* levels were significantly upregulated in the different muscle groups of the Huai pigs, whereas *CXCL10*, *ACTC1*, *MYH10*, and *SPTLC3* levels were significantly downregulated. In addition, the expression levels of the same DEG differed across muscle groups within the same breed. *CXCL10* and *MYH10* are closely associated with muscle growth and development [41,42]. *ACTC1* is a key gene involved in early myogenesis that also plays an important role in the growth and development of embryonic muscles; its low expression is conducive to muscle growth and development [43,44]. *CSRP3* and *LMOD2* play crucial roles in muscle fiber composition, can affect the distribution of muscle fiber types as well as meat quality, and have important positive regulatory effects [45,46]. *SPTLC3* and *NR3C2* play regulatory roles in lipid metabolism [47,48]. Our results showed that Huai pork was of higher quality than Duroc pork, based on six key regulatory genes identified to be related to pork quality traits and muscle development.

In this study, KEGG pathway analysis was used to assess the DEGs enriched in different muscle groups of Huai and Duroc pigs. The significantly enriched pathways were mainly associated with amino acid metabolism, PPAR signaling pathway, cAMP signaling pathway, and ECM–receptor interaction. The PPAR signaling pathway plays an important role in carbohydrate and lipid metabolism as well as muscle development and growth [49]. Alternatively, cAMP signaling promotes lipid metabolism and differentiation [50]. ECM–receptor interactions form a network with pathways related to lipid metabolism, thereby affecting IMF deposition [51,52]. In our analysis, Huai pigs exhibited higher expression levels of *PIM1* and *FABP3* and lower expression levels of *OLR1*, *TNNT2*, and *THBS1* than those observed in Duroc pigs. *PIM1* actively regulates myoblast function and skeletal muscle regeneration [53]. *FABP3* and *OLR1* play important roles in fat metabolism [54,55]. *TNNT2* is positively correlated with IMF content [56]. *THBS1* plays an important role in animal muscle growth and development [57]. These differences might have led to increased muscle mass in the Huai pigs compared to that observed in Duroc pigs.

## 5. Conclusions

Significant differences in meat quality and muscle fiber content were identified between different muscle groups of Huai and Duroc pigs. The proportion of oxidized muscle fibers in the different muscle groups of Huai pigs was significantly higher than that in Duroc pigs. The proportion of oxidized muscle fibers in the PM was significantly higher than that in the LD and BF muscles, whereas the IMF content in the PM was significantly higher than that in the LD and BF muscles. The differences in muscle fiber type and IMF content in muscles from Huai and Duroc pigs might impact meat quality traits, as reflected by the demand for different pork products and the need to improve pork yield.

In this study, the transcriptomes of LD, PM, and BF of Huai and Duroc pigs were comprehensively assessed. There were significant differences in meat quality, muscle fiber content, and IMF content between the two breeds. Several candidate genes (*CSRP3*, *LMOD2*, *NR3C2*, *CXCL10*, *ACTC1*, *MYH10*, *SPTLC3*) related to meat quality traits, muscle fibers, and IMF were identified. Our findings provide new insights into the regulatory mechanism of different meat quality traits associated with Huai and Duroc pigs and a theoretical reference for marker-assisted selection breeding based on DEGs to improve pork yield and quality in later stages.

## Figures and Tables

**Figure 1 genes-14-01548-f001:**
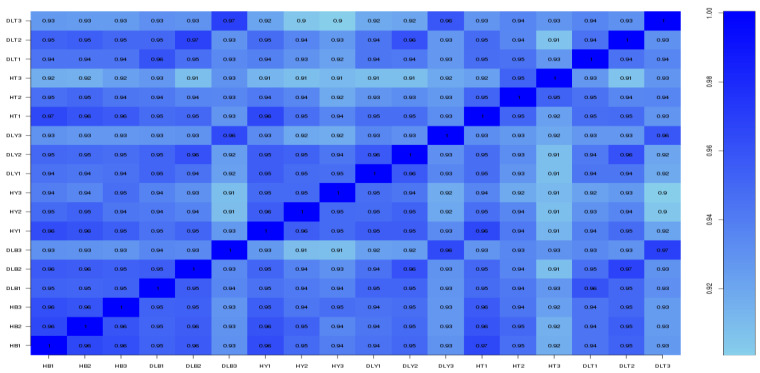
Heatmap of gene expression correlation.

**Figure 2 genes-14-01548-f002:**
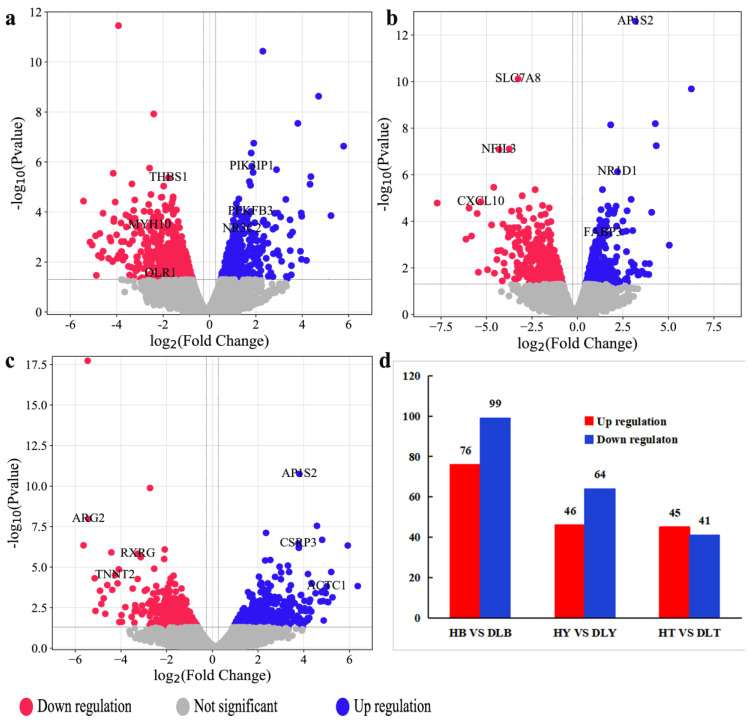
Volcano scatterplot and the number of DEGs. (**a**) Volcano scatterplot of HB vs. DLB. (**b**) Volcano scatterplot of HY vs. DLY. (**c**) Volcano scatterplot of HT vs. DLT. (**d**) Number of DEGs between HB vs. DLB, HY vs. DLY, and HT vs. DLT.

**Figure 3 genes-14-01548-f003:**
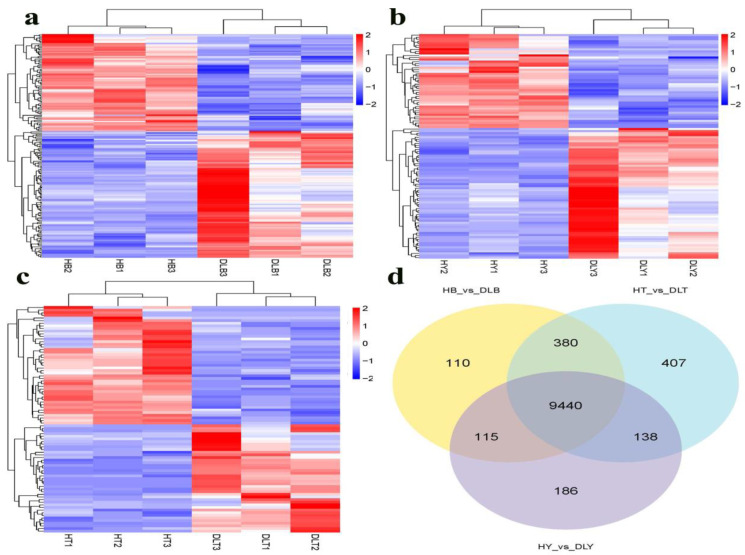
Heatmaps of DEG expression levels among the three comparison groups, representing three biological replicates. (**a**) HB vs. DLB. (**b**) HY vs. DLY. (**c**) HT vs. DLT. (**d**) Venn diagram of differential genes with an FPKM > 1.

**Figure 4 genes-14-01548-f004:**
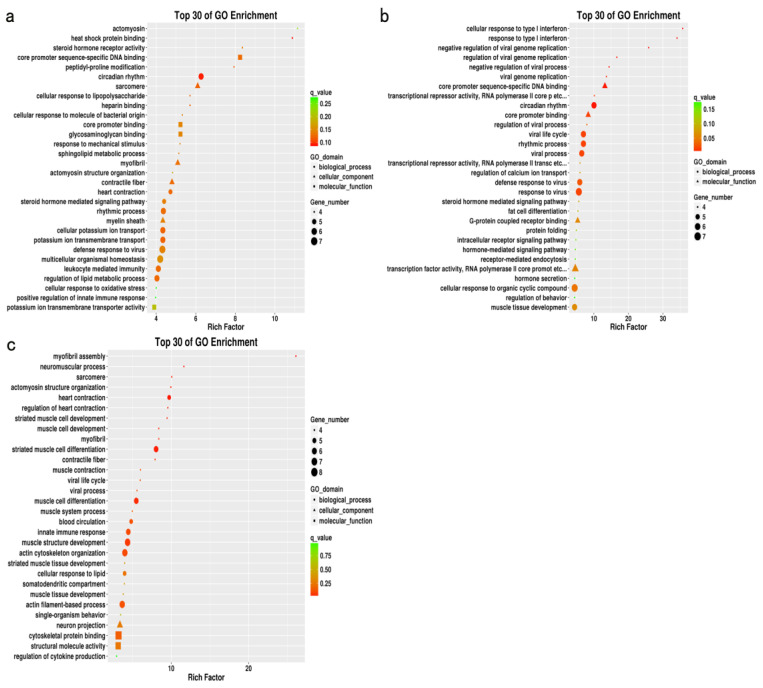
GO enrichment map of DEGs between HB vs. DLB, HY vs. DLY, and HT vs. DLT. (**a**) HB vs. DLB. (**b**) HY vs. DLY. (**c**) HT vs. DLT.

**Figure 5 genes-14-01548-f005:**
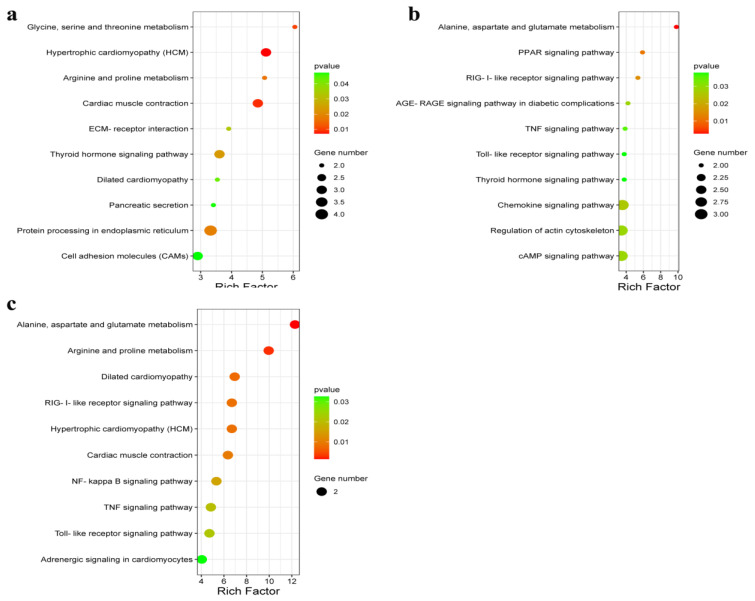
KEGG enrichment map of DEGs between HB vs. DLB, HY vs. DLY, and HT vs. DLT. (**a**) HB vs. DLB. (**b**) HY vs. DLY. (**c**) HT vs. DLT.

**Figure 6 genes-14-01548-f006:**
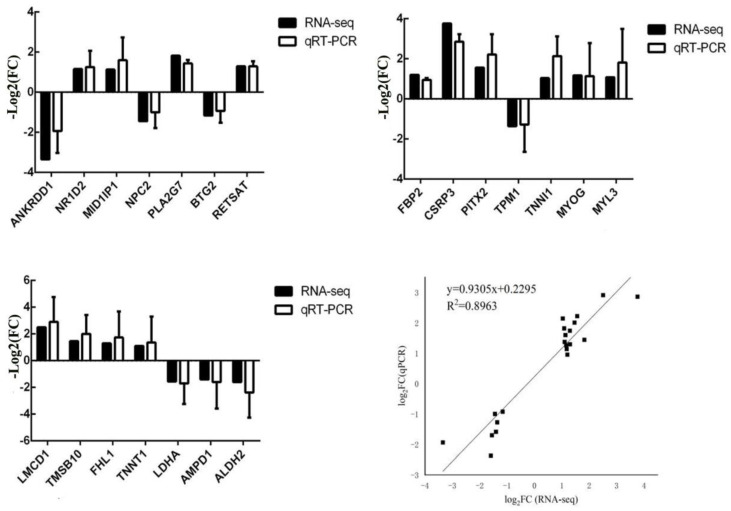
qPCR verification results of DEGs in different muscle tissues of different pig breeds.

**Table 1 genes-14-01548-t001:** Comparison of muscle mass traits of LD muscle.

Item	Huai Pig	Duroc Pigs
pH45min	6.20 ± 0.21	6.36 ± 0.31
L*	39.04 ± 0.40 ^A^	41.70 ± 2.12 ^B^
a*	10.15 ± 0.50 ^A^	8.35 ± 1.39 ^B^
b*	2.43 ± 0.18	2.72 ± 0.61
Water-holding capacity (%)	52.96 ± 1.34 ^A^	43.80 ± 2.44 ^B^
Water content (%)	73.81 ± 0.55 ^a^	72.26 ± 0.20 ^b^
IMF (%)	4.50 ± 0.13 ^A^	2.90 ± 0.07 ^B^
Drip loss (%)	1.44 ± 0.12 ^A^	2.01 ± 0.20 ^B^
Cooking loss (%)	54.12 ± 1.07 ^A^	59.39 ± 1.08 ^B^
Hardness (g)	505.67 ± 39.20 ^a^	585.14 ± 28.40 ^b^
Elasticity (mm)	5.60 ± 0.27 ^A^	3.12 ± 0.36 ^B^

Note: different lowercase letters (a/b) indicate significant differences (*p* < 0.05), whereas different uppercase letters (A/B) indicate *p* < 0.01.

**Table 2 genes-14-01548-t002:** Comparison of muscle mass traits of PM muscle.

Item	Huai Pig	Duroc Pigs
pH45min	6.02 ± 0.07	5.91 ± 0.13
L*	39.48 ± 0.92 ^A^	42.86 ± 0.93 ^B^
a*	14.83 ± 1.01 ^a^	13.48 ± 1.30 ^b^
b*	2.81 ± 0.44	3.27 ± 0.75
Water-holding capacity (%)	50.36 ± 2.35 ^A^	41.01 ± 1.56 ^B^
Water content (%)	75.64 ± 0.36	75.47 ± 0.30
IMF (%)	2.35 ± 0.06 ^A^	1.68 ± 0.21 ^B^
Drip loss (%)	2.05 ± 0.18 ^A^	2.61 ± 0.17 ^B^
Cooking loss (%)	55.9 ± 0.63 ^A^	58.48 ± 0.85 ^B^
Hardness (g)	435.78 ± 35.01 ^A^	547.50 ± 26.99 ^B^
Elasticity (mm)	5.69 ± 0.18 ^A^	4.54 ± 0.32 ^B^

Note: different lowercase letters (a/b) indicate significant differences (*p* < 0.05), whereas different uppercase letters (A/B) indicate *p* < 0.01.

**Table 3 genes-14-01548-t003:** Comparison of muscle mass traits of BF muscle.

Item	Huai Pig	Duroc Pigs
pH45min	6.40 ± 0.19	6.65 ± 0.25
L*	38.19 ± 0.67 ^A^	42.94 ± 1.81 ^B^
a*	14.95 ± 0.91 ^A^	8.62 ± 0.89 ^B^
b*	2.52 ± 0.08	2.77 ± 0.28
Water-holding capacity (%)	55.46 ± 1.99 ^A^	51.26 ± 5.23 ^B^
Water content (%)	74.76 ± 0.61 ^a^	73.09 ± 1.49 ^b^
IMF (%)	1.94 ± 0.01 ^a^	1.83 ± 0.29 ^b^
Drip loss (%)	2.79 ± 0.32 ^A^	3.08 ± 0.28 ^B^
Cooking loss (%)	57.49 ± 2.20 ^A^	59.53 ± 2.61 ^B^
Hardness (g)	681.11 ± 27.10 ^A^	936.56 ± 28.63 ^B^
Elasticity (mm)	6.98 ± 0.32 ^A^	3.37 ± 0.21 ^B^

Note: different lowercase letters (a/b) indicate significant differences (*p* < 0.05), whereas different uppercase letters (A/B) indicate *p* < 0.01.

**Table 4 genes-14-01548-t004:** Amino acid composition and content of LD muscle.

Amino Acid	Huai Pig	Duroc Pigs
Valine (Val)	1.19 ± 0.08	1.12 ± 0.03
Isoleucine (Ile)	1.08 ± 0.06	1.04 ± 0.03
Leucine (Leu)	2.02 ± 0.12	1.92 ± 0.03
Phenylalanine (Phe*)	0.88 ± 0.05	0.86 ± 0.02
Threonine (Thr)	1.13 ± 0.01	1.12 ± 0.04
Lysine (Lys)	2.28 ± 0.12	2.11 ± 0.06
Methionine (Met)	0.55 ± 0.01	0.60 ± 0.03
Aspartic acid (Asn*)	2.46 ± 0.06	2.27 ± 0.05
Glutamic acid (Glu*)	4.26 ± 0.05	4.08 ± 0.10
Proline (Pro)	0.78 ± 0.02	0.74 ± 0.02
Glycine (Gly*)	1.04 ± 0.08	0.99 ± 0.04
Alanine (Ala*)	1.34 ± 0.13	1.27 ± 0.03
Serine (Ser)	1.06 ± 0.08	1.01 ± 0.04
Tyrosine (Tyr*)	0.88 ± 0.03	0.79 ± 0.01
Histidine (His)	1.23 ± 0.10	1.15 ± 0.04
Arginine (Arg)	1.56 ± 0.04	1.45 ± 0.03
Delicious amino acids	12.67 ± 0.30 a	12.04 ± 0.23 b
TAA∑	24.64 ± 0.82	23.38 ± 0.38
EAA∑	9.14 ± 0.43 a	8.78 ± 0.20 b
NEAA∑	14.62 ± 0.34 a	13.75 ± 0.18 b

Note: Different lowercase letters indicate significant differences (*p* < 0.05). EAA = essential amino acid; EAA Σ = Thr + Val + Met + Ile + Leu; DAA = delicious amino acid; DAA Σ = Asn + Glu + Phe + Gly + Ala + Tyr; NEAA = nonessential amino acid; NEAA Σ = Asp + Cys + Tyr, TAA = total amino acid.

**Table 5 genes-14-01548-t005:** Amino acid composition and content of PF muscle.

Amino Acid	Huai Pig	Duroc Pigs
Valine (Val)	0.99 ± 0.05	1.04 ± 0.11
Isoleucine (Ile)	0.89 ± 0.05	1.01 ± 0.05
Leucine (Leu)	1.67 ± 0.04	1.77 ± 0.02
Phenylalanine (Phe*)	0.74 ± 0.05	0.81 ± 0.08
Threonine (Thr)	0.97 ± 0.04	1.04 ± 0.10
Lysine (Lys)	1.91 ± 0.06	2.03 ± 0.16
Methionine (Met)	0.49 ± 0.03	0.50 ± 0.03
Aspartic acid (Asn*)	2.19 ± 0.16	2.05 ± 0.17
Glutamic acid (Glu*)	3.82 ± 0.20	3.71 ± 0.05
Proline (Pro)	0.75 ± 0.01	0.71 ± 0.02
Glycine (Gly*)	0.91 ± 0.04	0.87 ± 0.07
Alanine (Ala*)	1.14 ± 0.10	1.24 ± 0.20
Serine (Ser)	0.88 ± 0.06	0.92 ± 0.11
Tyrosine (Tyr*)	0.79 ± 0.06	0.76 ± 0.09
Histidine (His)	0.97 ± 0.01	0.95 ± 0.06
Arginine (Arg)	1.30 ± 0.08	1.43 ± 0.08
Delicious amino acids	11.20 ± 0.19 a	11.07 ± 0.62 b
TAA∑	21.12 ± 0.54	21.65 ± 1.22
EAA∑	7.65 ± 0.30	8.20 ± 0.48
NEAA∑	12.73 ± 0.21	12.65 ± 0.66

Note: Different lowercase letters indicate significant differences (*p* < 0.05). EAA = essential amino acid; EAA Σ = Thr + Val + Met + Ile + Leu; DAA = delicious amino acid; DAA Σ = Asn + Glu + Phe + Gly + Ala + Tyr; NEAA = non-essential amino acid; NEAA Σ = Asp + Cys + Tyr, TAA = total amino acid.

**Table 6 genes-14-01548-t006:** Amino acid composition and content of BM muscle.

Amino Acid	Huai Pig	Duroc Pigs
Valine (Val)	1.09 ± 0.08	1.06 ± 0.05
Isoleucine (Ile)	0.98 ± 0.07	0.96 ± 0.05
Leucine (Leu)	1.89 ± 0.02	1.85 ± 0.05
Phenylalanine (Phe*)	0.84 ± 0.01	0.83 ± 0.01
Threonine (Thr)	1.03 ± 0.08	1.06 ± 0.06
Lysine (Lys)	2.07 ± 0.13	2.10 ± 0.05
Methionine (Met)	0.59 ± 0.01	0.53 ± 0.02
Aspartic acid (Asn*)	2.17 ± 0.46	1.95 ± 0.03
Glutamic acid (Glu*)	4.03 ± 0.06	3.82 ± 0.04
Proline (Pro)	0.76 ± 0.02	0.73 ± 0.01
Glycine (Gly*)	0.95 ± 0.08	0.95 ± 0.01
Alanine (Ala*)	1.34 ± 0.09	1.36 ± 0.12
Serine (Ser)	0.99 ± 0.04	0.95 ± 0.05
Tyrosine (Tyr*)	0.84 ± 0.02	0.74 ± 0.02
Histidine (His)	1.18 ± 0.06	1.07 ± 0.13
Arginine (Arg)	1.48 ± 0.05	1.42 ± 0.03
Delicious amino acids	11.91 ± 0.18 a	11.33 ± 0.14 b
TAA∑	23.08 ± 0.52	22.22 ± 0.45
EAA∑	8.50 ± 0.39	8.40 ± 0.24
NEAA∑	13.74 ± 0.21	12.99 ± 0.24

Note: Different lowercase letters indicate significant differences (*p* < 0.05). EAA = essential amino acid; EAA Σ = Thr + Val + Met + Ile + Leu; DAA = delicious amino acid; DAA Σ = Asn + Glu + Phe + Gly + Ala + Tyr; NEAA = non-essential amino acid; NEAA Σ = Asp + Cys + Tyr, TAA = total amino acid.

**Table 7 genes-14-01548-t007:** Summary statistics of the transcriptome.

Sample ID	Raw Reads	Clean Reads	Clean Ratio	Mapped Reads	Unique Reads	Mapping Ratio
DLB1	41,662,904	39,449,664	94.69%	32,,011,831	31,699,871	84.95%
DLB2	42,641,208	40,724,091	95.50%	33,460,379	33,143,647	85.18%
DLB3	42,844,548	40,865,654	95.38%	33,834,880	33,551,317	85.86%
DLT1	36,166,730	34,115,832	94.33%	28,102,269	27,842,623	85.83%
DLT2	32,862,880	30,900,715	94.03%	24,659,634	24,370,834	83.99%
DLT3	36,499,684	34,467,150	94.43%	28,158,235	27,911,410	85.56%
DLY1	36,215,306	34,496,888	95.25%	29,185,716	28,958,240	87.77%
DLY2	38,278,302	36,163,121	94.47%	29,563,660	29,269,444	85.66%
DLY3	36,475,654	34,576,215	94.79%	28,678,607	28,433,865	86.61%
HB1	32,091,128	30,439,453	94.85%	24,749,002	24,478,813	84.81%
HB2	35,156,424	33,446,968	95.14%	27,313,805	27,025,338	84.94%
HB3	38,263,956	36,283,053	94.82%	29,247,885	28,907,202	84.19%
HT1	33,392,590	31,752,953	95.09%	26,483,908	26,236,796	86.58%
HT2	37,773,530	35,619,382	94.30%	29,582,330	29,326,574	86.66%
HT3	42,016,924	39,639,585	94.34%	32,957,632	32,665,897	86.88%
HY1	40,655,380	38,870,213	95.61%	32,297,753	31,987,252	85.97%
HY2	37,437,228	35,804,116	95.64%	29,364,524	29,045,354	84.97%
HY3	38,432,316	36,425,629	94.78%	29,474,709	29,134,613	84.46%

## Data Availability

The original transcriptome sequencing data from the LD, BF, and PM muscles of Huai and Duroc pigs were uploaded to the NCBI database under the database entry number PRJNA962316.

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
