# Peer review of "Transcriptomic Profiling of Meat Quality Traits of Skeletal Muscles of the Chinese Indigenous Huai Pig and Duroc Pig"

_genes, 2023, doi:10.3390/genes14081548_

Round 1

Reviewer 1 Report

This is an interesting and well-written manuscript aimed to investigate the potential genetic mechanism of meat quality differences in different pig breeds or muscle parts. The introduction provides interesting information. Results are clear and discussion is well described and supported with several references. However, Conclusion and References sections should be improved. In addition, I suggest considering next minor grammar comments:

-       Line 30: Separate the round bracket from the text.

-       Lines 130 and 131: Replace “are” by “were”.

-       Line 162: Replace “Table2-4” by “Tables 2-4”.

-       Line 176: Replace “Table” by “Tables”.

-       Line 209: Replace “Figure” By “Figures”.

-       Line 246: Remove the comma.

-       Line 248: Comma signs appear to be displaced.

-       Line 251: Comma signs appear to be displaced.

-       Line 340: Separate the reference number from the text.

-       Lines 350-362: This section appears to be a brief summary of the results. I suggest including 3 or more conclusive sentences describing whether the results obtained were sufficient to meet the objective of the study, as well as possible implications and recommendations for future work.

-       Lines 383-502: There some mistakes in References section. Punctuation signs between author’s names are incorrect. Volume number should be italic. Volume number should be separated from the page range using a comma.

-       Line 383: Remove the period sign at the end of the Journal name.

-       Line 392: Remove the period sign at the end of the Journal name.

-       Line 398: Remove the period sign at the end of the Journal name.

-       Line 411: Remove the period sign at the end of the Journal name.

-       Line 446: Remove the period sign at the end of the Journal name.

-       Line 463: Remove the period sign at the end of the Journal name.

-       Line 482: Remove the period sign at the end of the Journal name.

Author Response

Comments and Suggestions for Authors

This is an interesting and well-written manuscript aimed to investigate the potential genetic mechanism of meat quality differences in different pig breeds or muscle parts. The introduction provides interesting information. Results are clear and discussion is well described and supported with several references. However, Conclusion and References sections should be improved. In addition, I suggest considering next minor grammar comments:

 1.Line 30: Separate the round bracket from the text.

Reply:Thank you very much for your suggestion. It has been revised in the newly submitted paper.

2.Lines 130 and 131: Replace “are” by “were”.

Reply:Thank you very much for your suggestion. It has been revised in the newly submitted paper.

Line 162: Replace “Table2-4” by “Tables 2-4”.

Reply:Thank you very much for your suggestion. It has been revised in the newly submitted paper.

Line 176: Replace “Table” by “Tables”.

Reply:Thank you very much for your suggestion. It has been revised in the newly submitted paper.

Line 209: Replace “Figure” By “Figures”.

Reply:Thank you very much for your suggestion. It has been revised in the newly submitted paper.

6.Line 246: Remove the comma.

Reply:Thank you very much for your suggestion. It has been revised in the newly submitted paper.

7.Line 248: Comma signs appear to be displaced.

Reply:Thank you very much for your suggestion. It has been revised in the newly submitted paper.

8.Line 251: Comma signs appear to be displaced.

Reply:Thank you very much for your suggestion. It has been revised in the newly submitted paper.     

9.Line 340: Separate the reference number from the text.

Reply:Thank you very much for your suggestion. It has been revised in the newly submitted paper.

10.Lines 350-362: This section appears to be a brief summary of the results. I suggest including 3 or more conclusive sentences describing whether the results obtained were sufficient to meet the objective of the study, as well as possible implications and recommendations for future work. 

Reply:Thank you very much for your suggestion. It has been revised in the newly submitted paper.

  1. Lines 383-502: There some mistakes in References section. Punctuation signs between author’s names are incorrect. Volume number should be italic. Volume number should be separated from the page range using a comma.

Reply:Thank you very much for your suggestion. It has been revised in the newly submitted paper.

12.Line 383: Remove the period sign at the end of the Journal name.

Reply:Thank you very much for your suggestion. It has been revised in the newly submitted paper.

13.Line 392: Remove the period sign at the end of the Journal name.

Reply:Thank you very much for your suggestion. It has been revised in the newly submitted paper.   

14.Line 398: Remove the period sign at the end of the Journal name.

Reply:Thank you very much for your suggestion. It has been revised in the newly submitted paper.      

15.Line 411: Remove the period sign at the end of the Journal name.

Reply:Thank you very much for your suggestion. It has been revised in the newly submitted paper.

16.Line 446: Remove the period sign at the end of the Journal name.

Reply:Thank you very much for your suggestion. It has been revised in the newly submitted paper.

Line 463: Remove the period sign at the end of the Journal name.

Reply:Thank you very much for your suggestion. It has been revised in the newly submitted paper.

18.Line 482: Remove the period sign at the end of the Journal name.

Reply:Thank you very much for your suggestion. It has been revised in the newly submitted paper.

Reviewer 2 Report

Suggestions, 

1)    Avoid acronyms in abstract. e.g., RNA seq, DEGs etc

2)    Extensive editing of English language required. At many instances, sentences need re-phrasing. 

3)    Line 109-116. The methodology of RNA-Seq is not clearly written, please re-write. 

4)    Line 118. What do you mean by joints?

5)    Line 123. Which pearl script? Be specific.

6)    Line 377. Quote the NCBI-SRA number here.

7)    Table 1 can be moved as supplementary file. 

8)    Table 2, 3, 4. Provide units (measurements) if necessary.

9)    Figure 2. Please name at least 4-5 most significant genes in each volcano plot.

10) Figure 2d. Please name y-axis.

11) Figure 4 is not clear. You can keep only 5 or 7 most significant terms. Provide high quality figure. Same applies to Figure 5.

Extensive editing of English language required. At many instances, sentences need re-phrasing.

Author Response

1.Avoid acronyms in abstract. e.g., RNA seq, DEGs etc.

Reply: Thank you very much for your suggestion. It has been changed in the article.

  1. Extensive editing of English language required. At many instances, sentences need re-phrasing. 

Reply: Thank you very much for your suggestion. It has been changed in the article.

3.Line 109-116. The methodology of RNA-Seq is not clearly written, please re-write. 

Reply: Thank you very much for your suggestion. It has been changed in the article.

4.Line 118. What do you mean by joints?

Reply:This refers to filtering out reads containing 5' and 3' end joints. It has been completed in the article.

5.Line 123. Which perl script? Be specific.

Reply: Thank you very much for your suggestion. This is mainly about calculating the FPKM value, because the FPKM value can measure the expression of each gene in the sample.

   FPKM =

Among them, transcription reads are the number of fragment reads covering the entire gene exon. Transcription length indicates the total length of gene exon. Total mapped reads in run indicates the total number of fragment reads in all mapped genomes of the sample.

6.Line 377. Quote the NCBI-SRA number here.

Reply: Thank you very much for your suggestion. All raw data has been uploaded to the NCBI database and the corresponding login number has been obtained. The login number information is described in the Materials section.

7.Table 1 can be moved as supplementary file. 

Reply: Thank you very much for your suggestion. It has been changed in the article.

8.Table 2, 3, 4. Provide units (measurements) if necessary.

Reply: Thank you very much for your suggestion. Already added in the table.

9.Figure 2. Please name at least 4-5 most significant genes in each volcano plot.

Reply: Thank you very much for your suggestion. It has been changed in the article.

10.Figure 2d. Please name y-axis.

Reply: Thank you very much for your suggestion. Already added in comments.

11.Figure 4 is not clear. You can keep only 5 or 7 most significant terms. Provide high quality figure. Same applies to Figure 5.

Reply: Thank you very much for your suggestion.The HD picture has been changed.

Round 2

Reviewer 2 Report

Authors has addressed few comments. Some of the specific comments were ignored for authors. English language is still concern. Manuscript needs one more round of revision. 

I have following suggestion, 

1)    Please check the quality of figure 5. You can keep top 5-10 enriched pathways only.

2)    Table 7. What do you mean by unique reads here? I don’t see that information in paper.

3)    Please move data availability statement to bottom.

4)    Comment 11 is not addressed properly.

5)    English language is not sufficiently improved. Editing of English language is needed.

English language is not sufficiently improved. Editing of English language is needed.

Author Response

1.Please check the quality of figure 5. You can keep top 5-10 enriched pathways only.

Reply:Thank you very much for your suggestion. It has been revised in the newly submitted paper.

2.Table 7. What do you mean by unique reads here? I don’t see that information in paper.

Reply:Thank you very much for your suggestion. unique reads: There is only one location-matched reads in the Sus scrofa reference genome (Sscrofa11.1).

3.Please move data availability statement to bottom.

Reply:Thank you very much for your suggestion. It has been revised in the newly submitted paper.

4.Comment 11 is not addressed properly.

Reply:Thank you very much for your suggestion. It has been revised in the newly submitted paper.

5.English language is not sufficiently improved. Editing of English language is needed.

Reply:Thank you very much for your suggestion. The language has been retouched.